# The Absence of Phasins PhbP2 and PhbP3 in *Azotobacter vinelandii* Determines the Growth and Poly-3-hydroxybutyrate Synthesis

**DOI:** 10.3390/polym16202897

**Published:** 2024-10-15

**Authors:** Claudia Aguirre-Zapata, Daniel Segura, Jessica Ruiz, Enrique Galindo, Andrés Pérez, Alvaro Díaz-Barrera, Carlos Peña

**Affiliations:** 1Departamento de Ingeniería Celular y Biocatálisis, Instituto de Biotecnología, Universidad Nacional Autónoma de México (UNAM), Av. Universidad 2001, Col. Chamilpa, Cuernavaca 62210, Morelos, Mexico; claudia.aguirre@ibt.unam.mx (C.A.-Z.); enrique.galindo@ibt.unam.mx (E.G.); 2Departamento de Microbiología Molecular, Instituto de Biotecnología, Universidad Nacional Autónoma de México (UNAM), Av. Universidad 2001, Col. Chamilpa, Cuernavaca 62210, Morelos, Mexico; daniel.segura@ibt.unam.mx (D.S.); jessica.ruiz@ibt.unam.mx (J.R.); 3Escuela de Ingeniería Bioquímica, Pontificia Universidad Católica de Valparaíso, Av. Brasil 2147 Casilla 4059, Valparaíso 2340025, Chile; aaperezb17@gmail.com (A.P.); alvaro.diaz@pucv.cl (A.D.-B.)

**Keywords:** poly-3-hydroxybutyrate, molecular mass, phasins, *Azotobacter vinelandii*

## Abstract

Phasins are proteins located on the surface of poly-3-hydroxybutyrate (P3HB) granules that affect the metabolism of the polymer, the size and number of the granules, and some also have stress-protecting and growth-promoting effects. This study evaluated the effect of inactivating two new phasins (PhbP2 or PhbP3) on the cellular growth, production, and molecular mass of P3HB in cultures under low or high oxygen transfer rates (OTR). The results revealed that under high OTRₘₐₓ conditions (between 8.1 and 8.9 mmol L^−1^ h^−1^), the absence of phasins PhbP2 and PhbP3 resulted in a strong negative effect on the growth rate; in contrast, the rates of specific oxygen consumption increased in both cases. This behavior was not observed under a low oxygen transfer rate (3.9 ± 0.71 mol L^−1^ h^−1^), where cellular growth and oxygen consumption were the same for the different strains evaluated. It was observed that at high OTR, the absence of PhbP3 affected the production of P3HB, decreasing it by 30% at the end of cultivation. In contrast, the molecular weight remained constant over time. In summary, the absence of phasin PhbP3 significantly impacted the growth rate and polymer synthesis, particularly at high maximum oxygen transfer rates (OTRₘₐₓ).

## 1. Introduction

Poly-3-hydroxybutyrate (P3HB) is a material recognized as a bioplastic material that belongs to the polyester family of polyhydroxyalkanoates (PHAs). P3HB shares thermomechanical properties with currently used petrochemical plastics, but it is biodegradable and biocompatible [1]. Therefore, P3HB is a material utilized in diverse areas, like biomedical, pharmaceutical, veterinary, food packaging, and cosmetics [2,3].

Many species of bacteria use different carbon sources to accumulate P3HB as a natural mechanism for carbon and energy storage [4]. *Azotobacter vinelandii* is an interesting P3HB production model, able to synthesize this polymer under conditions of nutritional limitation, either of phosphates or mainly of oxygen and using different substrates, such as cane molasses, glucose, sucrose, and compostable waste [5].

P3HB is stored in granules, also named “carbonosomes” due to their complexity acquired through the binding of diverse GAPs (granule-associated proteins), which are P3HB synthases, P3HB depolymerases, regulatory proteins (like PhaR), and phasins [6,7]. Phasins are non-catalytic proteins (10–24 kDa), and are the major group of 2GAPs on the P3HB granule surface. Phasins are very important in controlling the size, number, and distribution of the granules in the cells, in addition to regulating the P3HB synthase and P3HB depolymerase activities [7,8,9].

It has been observed that the overexpression of some phasin genes is related to increased accumulation of P3HB [8,9,10,11]. Likewise, it is known that phasins can also contribute to the regulation of P3HB depolymerization. Phasin PhaP could function as an activator of the soluble PHB depolymerase of *R. rubrum*; surprisingly, PhaP is also a thermoresistant and chemoresistant protein [10]. On the other hand, it was suggested that in *Cupriavidus necator*, phasins modulate the activity of P3HB depolymerase Z1 and Z2, respectively [12].

One of the hypotheses that explains the relationship between phasins and the modulation of the activity of synthases and depolymerases is that, since the phasins are found on the surface of the granules, they interact with P3HB synthases and P3HB depolymerases, causing allosteric changes that modify the activity or binding to the PHB granule [13,14].

On the other hand, it has been proposed that phasins can act as chaperones that help the cell relieve different types of stresses, such as thermal shocks, halophilic environments, solvents and oxidative stress [7,15,16].

*A. vinelandii* is a P3HB producer bacterium. Three different phasin proteins have been identified in this microorganism. The major protein in the P3HB granules of this bacterium is PhbP1. This protein is involved in the determination of size and number of granules [17]. The second phasin PhbP2 [18], belongs to the same family of phasins as PhbP1. Both proteins have at the same position, a domain Phasin_2 (pfam09361; TIGR01841) that is characteristic of this group of phasins. PhbP1 and PhbP2 phasins are similar in length (186 aa, Mw 20422.82; 179 aa, Mw 19187.69, respectively), although their amino acid sequences are only 31.4% identical. With respect to PhbP3, this protein is also small (203 aa, Theoretical Mw 21789.22), and is also present on the P3HB granules [17]. PhbP3 is not homologous to PhbP1 and PhbP2 (only 15.7% identical to PhbP1 and 20.8% to PhbP2) and it has no significant homology either with other proteins recognized as phasins in other organisms. No conserved domains could be found in its sequence and clear orthologous proteins were found only in *Azotobacter* spp, where they are annotated as hypothetical. However, it is interesting to note that the three phasins have hydrophobic regions in their sequence, as other phasin proteins have, that could be involved in their binding to the P3HB granules. Although their sequences are not very similar, they share some structural characteristics, like regions predominantly of alpha-helical arrangements, with some regions predicted to be disordered or unstructured, which is another characteristic of several phasins. This structural flexibility suggests that it could interact with different targets, including misfolded proteins [15]. It is known that the inactivation of PhbP2 phasin promotes the synthesis of P3HB, and that inactivation of PhbP3 phasin reduces P3HB production in *A. vinelandii* in cultures in shaken flasks in PYS-rich medium [19,20]. However, the mechanism by which PhbP2 and PhbP3 phasins are involved in the regulation of P3HB synthesis and degradation has not been described in detail.

On the other hand, it has been reported that the increase in the agitation rate in bioreactors, and therefore in OTR, can generate an effect known as “sublethal damage”, where it is proposed that the high OTR results in a condition of oxidative stress [21,22]. OTR is an important factor in the operation and optimization of biopolymer production. In microorganisms like *A. vinelandii,* a thorough understanding of OTR and culture under oxygen-limited conditions is crucial for P3HB production [23].

Taking the above into consideration, the present study aimed to evaluate the effect of inactivating phasins PhbP2 or PhbP3 on the growth rate, production, and molecular mass of P3HB in cultures at low and high OTR, to better understand their role in the P3HB metabolism of *A. vinelandii* and to contribute to the design of strains capable of improving specific characteristics, such as molecular size.

## 2. Materials and Methods

### 2.1. Maintenance and Preservation of Strain

*A. vinelandii* strain OP, which is non-mucoid, was used as wild type. Strains OP-PhbP2^−^ and OP-PhbP3^−^ are mutant derivatives of strain OP containing gene inactivations of the *phbP2* and *phbP3* phasin genes, respectively, and were constructed as follows. In mutant OP-PhbP2^−^, the gene coding for the phasin named PhbP2 (gene *avin_03930*) was inactivated by insertion of a tetracycline resistance cassette derived from plasmid pHP45 Ω-Tc [24] into the BstXI restriction site within its coding sequence. This mutant allele was introduced into the chromosome of strain OP by double-crossover recombination and selection of tetracycline resistance (genotype *phbP2::Tc^R^*). In mutant strain OP-PhbP3^−^ the gene coding for a different phasin named PhbP3 (gene *phbP3*; *avin34720*) was inactivated. This was accomplished by inserting a kanamycin resistance cassette from plasmid pBSL99-Km [25] into the HincII restriction site found in the *phbP3* gene. The resulting mutant allele was introduced into the chromosome of strain OP by gene replacement through a double-crossover recombination and selecting on kanamycin (genotype *phbP3::Km^R^*). The introduction of each gene inactivation was confirmed by PCR and produced truncated phasins [20]. The strains were stored in cryovials with 40% glycerol at −70 °C.

The OP, OP-PhbP2^−^, and OP-PhbP3^−^ strains were grown in Petri dishes with solid PYS medium, which contains sucrose (20 g L^−1^), peptone (5 g L^−1^), yeast extract (3 g L^−1^), agar (18 g L^−1^) and the corresponding antibiotics selection, tetracycline (60 μg mL^−1^) and kanamycin (3 μg mL^−1^). The inoculum was obtained through cells grown in a 500 mL shaken flask with 100 mL of liquid PYS medium (in g L^−1^: 20 sucrose, 5 peptone and 3 yeast extract) without antibiotic, under 200 rpm and 29 °C, for 24 h until reaching an optical density of 0.16 ± 0.02 at 540 nm (Genesys 10S UV-VIS, Thermo Scientific, Boston, MA, USA), corresponding to 0.08 ± 0.02 g L^−1^ of cell dry weight.

### 2.2. Bioreactor Cultures

The culture was carried out in a 3 L Applikon (Delft, The Netherlands) bioreactor, equipped with two Rushton turbines (impeller diameter/tank diameter = 0.35), 6 flat-blades and a 7-hole diffuser to provide bubbling aeration. pH was controlled to 7.2 ± 0.1 by the automatic addition of NaOH 2N and HCl 2N through cultivation. Cultures were conducted in PYS medium, at 29 °C using an agitation rate of 300 and 500 rpm for all strains evaluated, with a working volume of 2 L and aeration of 1 vvm. A gas analyzer (Teledyne Analytical Instruments, City of Industry, CA, USA, model 7500) was used to measure O_2_ and CO_2_ in the gaseous flow at the bioreactor output. The gas analysis equipment was calibrated using nitrogen (auto zero setting) and a reference gas (1% CO_2_ and 5% O_2_ for SPAN).

The estimation of the oxygen OTR was made from the online analysis of the level of gaseous oxygen at the outlet of the bioreactor [21]:(1)$$OTR=\frac{M_{O^{2}}*F_{G}^{in}}{V_{R}*V_{M}}(X_{O_{2}}^{in}-X_{O_{2}}^{out})$$
where $M_{O_{2}}$ is the molecular mass of oxygen (g mmol^−1^), $F_{G}^{in}$ is the volumetric inlet air flow at standard conditions (L h^−1^), $V_{R}$ is the working volume (L), $V_{M}$ is the mol volume of the ideal gas at fraction of oxygen in the inlet air (mol mol^−1^), $X_{O_{2}}^{out}$ is the molar fraction of oxygen in the inlet air (mol mol^−1^), $X_{O_{2}}^{out}$ is the molar fraction of the oxygen in the outlet gas of the fermenter (mol mol^−1^).

### 2.3. Determination of Fermentation Parameters

The specific growth rate (*μ*) was calculated as described [26].

The *μ* was calculated at 0 to 24 h of the culture using the following equation:(2)$$\frac{dX}{dt}=\mu X$$
where *μ* is the specific growth rate (h^−1^) and *X* is the cellular protein concentration (g L^−1^).

The P3HB volumetric productivity was determined according to:(3)$$Q_{P3HB}=\frac{P-P0}{∆t}$$

*P*_0_ is the initial P3HB concentration (g L^−1^), *P* is the 3HB concentration (g L^−1^) to 24, 48 or 72 h, and ∆*t* is the time in the period.

The specific oxygen uptake rate (qO_2_) was calculated with the following equation:(4)$$qO_{2}=\frac{OTR\;(mmol\;L^{-1}h^{-1})}{cellular\;protein\;(gL^{-1})}$$

### 2.4. Biomass and P3HB Quantification

Bacterial growth was measured as protein and measured using the Lowry method [27]. To quantify P3HB, 3–5 mg of dry biomass was taken and 1 mL of concentrated H_2_SO_4_ was added. The mixture was heated to 90 °C for 1 h. The samples were allowed to cool, then diluted 1:50 with MilliQ water (MilliporeSigma, Burlington, MA, USA) and a 20 μL sample was injected into high-pressure liquid chromatography (HPLC) (Waters Alliance 2695) using an Aminex HPX-87H column (Bio Rad, Philadelphia, PA, USA), using 5 mM H_2_SO_4_ mobile phase, and a Waters 2996 diode array detector (Milford, MA, USA) was used [19]. The area under the curve was quantified at 220 nm. The standard was prepared through hydrolysis of commercial P3HB (1–0.1 mg mL^−1^) [23].

### 2.5. Molecular Mass Determination of P3HB

The biomass contained in 3 or 6 mL of sample was recovered by centrifugation. Subsequently, the biomass was washed with 1 mL of distilled water, resuspended, and centrifuged again at 8060× *g* for 10 min (Eppendorf, model 5804 R, Hamburg, Germany). A total of 1 mL of acetone was added, and the cell pack was agitated for 10 min. The sample was centrifuged at 8060× *g* for 10 min, acetone was discarded and 2 mL of chloroform was added to solubilize P3HB, leaving it in contact for 20–24 h at room temperature for subsequent filtration before molecular mass analysis.

The molecular mass distribution was determined by gel permeation chromatography. A Shodex K-807L column (Resonac, Tokyo, Japan) was used, which permits the analysis of samples with molecular mass from 1000 to 20,000 kDa. The column was coupled to HPLC equipment (Waters Alliance 2695, Milford, MA, USA) with a refractive index detector (Waters, 2414, Milford, MA, USA). The injection volume was 50 μL, at a working temperature of 30 °C and a run time of 30 min at a flow of 1 mL min^−1^ using chloroform as a mobile phase. Polystyrene standards were used to elaborate the calibration curve with molecular mass between 2.9 × 10^3^ and 5.6.0 × 10⁶ Da. Samples were prepared at a concentration of 1–2 mg mL^−1^ and dissolved 24 h before analysis. Each sample was filtered with glass syringes and chloroform-resistant PTFE membranes with a pore size of 0.45 μm (Merk, Millipore, San Francisco, CA, USA, No. SLCR033NB).

Empower Chromatography Data System (Waters) was used for the processing and quantification of the molecular weight and mean molecular mass (MMM) of the samples. From the calibration curve, an equation was obtained to estimate the molecular weight of P3HB depending on the elution volume [23].

The depolymerization rate (Da h^−1^) was calculated using the following equation:(5)$$Depolymerization\;rate=\frac{{MMM}_{72\;h-}{MMM}_{24\;h}}{48\;h}\;$$
where *MMM_72_* is the mean molecular mass (Da) at 72 h, and *MMM_24_* is the mean molecular mass (Da).

## 3. Results and Discussion

### 3.1. Growth Kinetics, OTR Profiles and qO_2_ in Bioreactor Cultures at 300 and 500 rpm with OP, OP-PhbP2^−^ and OP-PhbP3^−^ Strain

From the three different phasin proteins identified in *A. vinelandii*, a physiological role has been established only for the majority phasin PhbP1 [18]. However, the role of PhbP2 and PhbP3 has not been studied. To evaluate the effect of the absence of phasins PhbP2 and PhbP3 under possible oxidative stress and start understanding their role in P3HB metabolism, the OP wild-type strain and its derivative mutants OP-PhbP2^−^ (gene *phbP2* inactivated) and OP-PhbP3^−^ (gene *phbP3* inactivated) were tested under different OTR conditions, represented by two agitation rates (300 and 500 rpm). These correspond to low and high OTR. This was analyzed because some phasins have been shown to participate in stress protection due to their chaperone-like activity [15,16,28].

Figure 1a shows the bacterial growth kinetics, measured as protein, of the three strains grown at 500 rpm in PYS medium. In the cultures using the OP-PhbP2^−^ and OP-PhbP3^−^ strains, the cellular growth was lower compared to that of the OP strain. In the case of strain OP, the maximal protein concentration was 1.43 ± 0.04 g L^−1^, whereas the cultures with the mutant strains OP-PhbP2^−^ and OP-PhbP3^−^ reached 0.96 ± 0.02 g L^−1^ and 1.08 ± 0.15 g L^−1^, respectively. The μ was 0.05 ± 0.01 h^−1^ for OP-PhbP2^−^ and 0.04 ± 0.01 h^−1^ for OP-PhbP3^−^. In the case of the culture with the parental strain OP, the μ was higher (0.08 ± 0.02 h^−1^) (Table 1).

Although the μ was lower in the cultures conducted with the mutant strains, the OTR profiles (Figure 1b) were very similar for the three strains evaluated. For example, the OTR increased during the first 10 h of cultivation, reaching maximal values between 8.1 and 8.9 mmol L^−1^ h^−1^ depending on the strain evaluated. Regardless of the strain, the OTR remained constant from 10 to 24 h of cultivation, which was a sign of oxygen limitation in the culture, with a plateau region characteristic of this kind of limitation, as previously described [21,29]. After 24 h, the OTR dropped drastically to a minimal value of 3 mmol L^−1^ h^−1^.

Figure 1c shows the sucrose during cultivation. For the three strains evaluated, the rate of sucrose consumption at 500 rpm was the same, with values of 0.25 g L^−1^ h. At the end of the culture (72 h), a residual sugar concentration of 2 g L^−1^ was determined for the cultures with the OP strain and OP-PhbP3^−^ strains, and 4.0 g L^−1^ in the cultures with the mutant OP-PhbP2^−^. It is known that in cultures of *A. vinelandii,* the affinity constant (K_s_) for sucrose is 0.1 g L^−1^ [30]. Therefore, the cultures were not limited by the carbon source, and thus, it is possible that other nutrients, such as phosphates or trace elements, are responsible for this drop in oxygen transfer rate and oxygen consumption.

The qO_2_ was calculated in the range of 12 to 36 h, since in that period, the dissolved oxygen tension remained constant, and therefore, it can be assumed that the oxygen transfer rate is equal to the oxygen uptake rate [30]. As shown in Figure 1d, the highest value of qO_2_ was reached with the mutant strains along the culture. In the cultures at 500 rpm, this value was two- to three-fold higher than that obtained with the OP strain during the growth phase (12–24 h). For example, at 12 h of culture, the qO_2_ for the culture of the OP-PhbP3^−^ strain was 37.3 ± 1.6 mmoL g^−1^ h^−1^ and 25.5 ± 1.0 mmoL g^−1^ h^−1^ for the culture with mutant OP-PhbP2^−^, whereas for the OP strain, it was 9.2 ± 1.0 mmoL g^−1^ h^−1^ (Table 1).

It has been reported that variations in agitation rate, and therefore in the OTR, could promote an effect known as “semi-lethal damage”, where it is proposed that high OTR, similar to those achieved at 500 rpm, results in a condition of oxidative stress [21,22]. On the other hand, it is known that *A. vinelandii* cell walls exhibit a high resistance to mechanical stress; therefore, it seems unlikely that cells of *A. vinelandii* could be damaged by the mechanical stress generated in the bioreactor at 500 rpm [31]. Previous studies have reported that in cultures of *A. vinelandii* at 500 rpm, it is possible to find a situation that causes semi-lethal cell damage [21]. Considering this starting point, and that in the mutant strains the phasins PhbP2 or PhbP3 are not expressed, we propose that these phasins may be playing another function distinct to P3HB metabolism, possibly as chaperone proteins that protect the cells against oxidative stress, as has been shown for other phasins [15,16]. To evaluate this possibility, cultures were carried out at low OTR (300 rpm). As shown in Figure 2a, there were no significant changes in the growth or the qO_2_, that being the case for the two mutant strains concerning strain OP. The values of maximal protein concentration were in the range of 0.9 to 1.4 g L^−1^, with μ 0.025 ± 0.001 h^−1^ for both OP and OP-PhbP2^−^ and 0.027 ± 0.001 h^−1^ for OP-PhbP3^−^. In the case of qO_2_, the values at 12 h of cultivation were in the range of 6.9 to 12.1 mmoL g^−1^ h^−1^ (Table 1). Therefore, it is possible that under conditions of high oxygen transfer, phasins could be involved in participating as chaperone-type proteins that help face a situation of oxidative stress in the cells. However, more studies are necessary to elucidate the mechanisms involved.

It is known that the phasins exert a stress reduction action, both in P3HB and non-P3HB-synthesizing bacteria, decreasing the induction of heat shock-related genes in *E. coli* [12,13] and promoting protein folding through a chaperone-like mechanism, which suggests an in vivo general protective role [28]. However, it is necessary to carry out more studies that help discern if the suggested effect is carried out genetically or if it is through protein–protein interactions on the granule surface, since in some models, it has been shown that phasins facilitate the anchoring of P3HB synthases to the surface of the granules [28].

### 3.2. P3HB Production at 300 and 500 rpm Using the OP and Mutant Strains

The accumulation percentage of P3HB and its concentration were also quantified at different culture times. Figure 3a shows the kinetics of P3HB production of the three strains evaluated at 500 rpm. It was observed that the absence of phasin PhbP2 increased the production of P3HB, whereas the absence of phasin PhbP3 resulted in a decrease in the production of the polymer. The maximal concentrations were 4.6 ± 0.5 g L^−1^ for the OP strain, 5.3 ± 0.3 g L^−1^ for the OP-PhbP2^−^ strain, and 3.7 ± 0.6 g L^−1^ for the OP-PhbP3^−^ strain (Table 2).

Figure 3c shows the accumulation percentage of P3HB at 500 rpm. In general, no significant differences were found in the P3HB accumulation in the different strains evaluated. The OP-PhbP2^−^ strain showed high levels of P3HB accumulation, reaching values of 82.3 ± 8.1% based on the dry weight of the bacteria. On the other hand, in the cultures with the OP strain, the percentage was 72.0 ± 0.5 w w^−1^, and 71.0 ± 0.5% w w^−1^ for the OP-PhbP3^−^ strain (Table 2).

It is important to point out that in the cultures at 300 rpm (OTR about 5 mmol L^−1^ h^−1^), both the concentration (Figure 3b) and the percentage of polymer accumulation (Figure 3d) were lower with the three strains evaluated than those obtained at 500 rpm. As shown in Figure 3b, the highest production of P3HB was obtained in the cultures with the OP strain, reaching 3.8 ± 0.6 g L^−1^. In the case the OP-PhbP3^−^ mutant strain, the P3HB production was 1.6 ± 0.3 g L^−1^, and for the OP-PhbP2^−^ strain, it was 1.4 ± 0.1 g L^−1^. Similarly, the maximal percentage of polymer accumulation was around 70% with the OP strain and 47.3 ± 1.0% w w^−1^ and 62.7 ± 3.0% w w^−1^ with OP-PhbP2^−^ and OP-PhbP3^−^, respectively. It is important to point out that at 300 rpm, in the culture with the mutant strains, a decrease in P3HB accumulation according to the culture time was observed. Although we do not have an explanation for this, it is possible that in the mutant strains, there is a slight consumption of P3HB from the beginning of cultivation, which is not necessarily reflected in the concentration of the polymer. It is important to note that the higher Q_P3HB_ was obtained in the cultures conducted at 500 rpm using the OP-PhbP2^−^ strain (Table 2).

These results contrast with previous reports by other authors, who found that under conditions of low oxygen transfer, such as those obtained at 300 rpm, the synthesis of P3HB is favored, because the carbon source is channeled through the TCA cycle, and much of the carbon source is directed to P3HB production instead of bacterial growth. However, those studies were performed with a different strain (OPNA) that has inactivated the *rsmA* and *ptsN* genes that are involved in the regulation of P3HB synthesis [30].

More recent studies [23] reported a behavior similar to that observed in this study. Those authors found that maximal production (4.2 ± 0.4 g L^−1^) and accumulation of P3HB (90 ± 5% w w^−1^) using the strain OP was reached in the cultures grown at 500 rpm (high OTR_max_). On the contrary, the lowest P3HB production of approximately 1.6 ± 0.3 g L^−1^ (56 ± 2% w w^−1^) was obtained at 300 rpm (low OTR_max_).

This increase in the production and accumulation of P3HB at high oxygen transfer may be due to the activation of a cell protection mechanism against oxidative stress, as was previously reported [22]. Those authors found that in cultures at high oxygen transfer (20.2 mmol L^−1^ h^−1^), *Rhizobium phaseoli* increases both the activity of catalase, an enzyme that acts on hydrogen peroxide, and the production of P3HB, as a strategy to address oxygen stress situations.

On the other hand, it was suggested that in *Cupriavidus necator*, phasins modulate the activity of P3HB depolymerases Z1 and Z2, respectively [32]. In the case of *Azotobacter* sp. FA-8, the phasin PhaPAz is the most abundant P3HB granule-associated protein [16,33]. It was found that this protein displays a growth-promoting effect, also enhancing the polymer production in recombinant P3HB-producing *Escherichia coli* [15].

### 3.3. Molecular Mass Distributions of Polymers Produced by the OP and Mutant Strains

Figure 4 shows the mean molecular mass data of the three strains evaluated at 500 rpm. Maximal molecular masses of 7080 ± 273 kDa and 6820 ± 237 kDa were reached in the P3HB produced by the OP strain and OP-PhbP2^−^ at 24 h of cultivation. At the end of the culture (72 h), the mean molecular mass of the polymer decreased in both strains to values of 5480 ± 239 kDa and 4489 ± 281 kDa for the polymers produced, respectively, by the OP strain and OP-PhbP2^−^. It is interesting to note that in the case of the strain OP-PhbP3^−^, the mean molecular mass remains constant throughout the cultivation at a value of 7262 ± 1334 kDa at 24 h of cultivation and 6704 ± 981 kDa at 72 h. According to the ANOVA analysis, there are significant differences (α < 0.05) in the MMM of the polymer produced by the OP and OP-PhbP2^−^ in the range of 24 to 72 h, whereas no significant differences were found in the MMM of the P3HB produced by the OP-PhbP3^−^ in the same period of cultivation (Figure 4).

P3HB with different molecular masses exhibits varying mechanical and processing properties, determining their applicability in commercial sectors [2]. Low-molecular-mass P3HB is preferred in applications requiring higher biodegradability and flexibility, such as single-use packaging, controlled drug release systems, and biodegradable sutures. In contrast, higher-molecular-mass P3HB is more suitable for products that demand high strength and durability, such as bioplastics for structural components, textiles, and implantable medical devices [2].

To better understand the changes observed in the molecular mass of P3HB, Figure 5 shows the molecular mass distribution of the P3HB accumulated at 24, 48, and 72 h for the three strains tested.

It is clear that in the case of the OP strain, at 24 h, most of the P3HB fractions are in the range of 1000 to 10,000 kDa, whereas at 48 and 72 h, a significant change in the distribution is found, with a high percentage of molecules in the range of 100 to 1000 kDa. On the other hand, in the polymer synthesized by the OP-PhbP2^−^ strain, a phenomenon similar to that identified in the OP strain is observed, especially at 72 h, where the percentage of molecules in the range of 100 to 1000 kDa increased significantly. In contrast, in the case of the OP-PhbP3^−^ strain, no relevant changes were observed in the distribution of molecular mass as a function of culture time and most of the fractions are in the range of 1000 to 10,000 kDa, independently of culture time. This behavior is reflected in the polydispersity index (PI) of the product, finding that in the polymer produced by the OP strain and OP-PhbP2^−^, the values at the end of the culture are higher than 3 (3.01 ± 2.04 and 4.93 ± 0.88, respectively), whereas for the polymer synthesized by the strain OP-PhbP3^−^, the PI was 1.27 ± 0.05.

The decrease in the molecular mass throughout the culture observed in the OP and OP-PhbP2^−^ strains could be related to the activity of P3HB depolymerase during the stationary phase of the cultivation. In the cultures with the OP and OP-PhbP2^−^ strains, activities of 1.20 ± 0.10 and 1.86 ± 0.20 μg P3HB mg prot^−1^ min^−1^ were detected, respectively, whereas in the OP-PhbP3^−^ strain, no activity was detected. Previous studies with the OP strain [17] have found an increase between 40 and 50% in the activity of P3HB depolymerase at the end of the exponential growth phase and the stationary phase, under oxygen transfer conditions similar to those used in the present study. In contrast, the P3HB synthase activity decreased by about 50% at the end of the stationary phase [34].

Regarding the results observed in the P3HB synthesized by the OP-PhbP3^−^ strain, it is possible to hypothesize that the phasin could activate the P3HB depolymerase PhbZ1, which is known to cause a decrease in the molecular mass of the P3HB [17,34]. The absence of phasin PhbP3 would negatively affect the activity of this depolymerase, so less hydrolysis of the polymer would occur, because a similar result to that of mutant OP-PhbP3^−^ has been observed with the inactivation of PhbZ1, maintaining the MMM along the culture [17]. This kind of effect of phasins on the activity of P3HB depolymerases has been previously reported [13,32].

There are some examples where changes in the expression of different phasins had effects on the molecular mass of the PHAs produced. For example, in *Aeromonas hydrophila,* the over-expression of the phasin PhaPAh reduced the molecular mass of the PHA produced to approximately 50% of that of the wild-type strain. This phenotype was explained by the authors as a possible indirect effect through the PHA synthase, because over-expression of *phaPAh* increased transcription of PhaCAh, and over-expression of *phaC* also negatively affected the molecular mass [35].

On the other hand, the phasin PhaPAh from *Aeromonas hydrophila* seems to regulate the *phaCAh* gene at the transcriptional level [35]. If a similar regulation occurs in *A. vinelandii,* the amount of PhbC would be affected in the phasin mutants, and a correlation between the active synthase concentration and the molecular mass of the PHA produced has been reported, where it was found that the lower the PhaC concentration, the higher the molecular mass of the polymer [36].

Table 3 shows the depolymerization rate of P3HB and Q_P3HB_ in cultures conducted at 500 rpm. It is possible to observe a direct relationship (r^2^ = 0.999) between the depolymerization rate (determined between 24 and 72 h) and the Q_P3HB_. For the lowest Q_P3HB_ (0.071 g L^−1^ h^−1^), obtained in the cultures at 500 rpm using the OP-PhbP3^−^ mutant strain, a depolymerization rate of 3555 ± 136 Da h^−1^ was reached. On the other hand, in the cultures with the OP strain, the Q_P3HB_ was 0.087 g L^−1^ h^−1^, with a depolymerization rate 5 times higher (15800 Da h^−1^) than that with the OP-PhbP3^−^ strain. The extreme condition occurs with the OP-PhbP2^−1^ strain, where the highest rate of P3HB synthesis was reached (0.098 g L^−1^ h^−1^), as well as a high rate of depolymerization (24500 Da h^−1^).

This trend was previously reported by [34], who found a close relationship between the synthesis rate of P3HB and the molecular mass of polymers. These authors reported that when the rate of P3HB synthesis is increased, by manipulating the polymer content in the inoculum, a significant decrease in the molecular mass of the polymer was observed. In addition, this behavior is similar to that observed in recombinant *E. coli* and *Azohydromonas lata* cultures, where an inverse relationship between the P3HB production rate and the molecular mass was reported [17,35,37]. On the other hand, studies in our group [32] have shown that in the case of the cultures developed using an inoculum with 50% of P3HB, the synthase activity was higher when the P3HB productivity was higher (growth exponential phase) and this activity decreases significantly during the stationary phase of growth when the productivity of P3HB decreases [33].

## 4. Conclusions

Overall, our results demonstrate for the first time that the absence of phasins PhbP2 or PhbP3 leads to a significant reduction in the specific growth rate of mutant strains under high OTR conditions. This behavior could be related to the protective role that some of these proteins have against the effect of oxidative stress generated in cultures under high oxygen transfer. For example, the phasin PhaP_Az_ from *Azotobacter* sp. FA-8, which belongs to the same family of phasins as PhbP2 of *A. vinelandii*, has a stress-reducing action that results in increased growth and higher resistance to superoxide stress. This phasin has chaperone-like activity, so the stress-protective effect could be related with this capacity [15,16]. The PhbP2 phasin could have a similar role, because both PhbP1 and PhbP2 of *A. vinelandii* have a similar predicted structure with several alpha-helical arrangements, alternated with unstructured regions. The regions of predicted structural flexibility could interact with different targets, including misfolded proteins. In the case of PhbP3, it represents a new family of phasins, with no conserved domains shared with PhbP1 and PhbP2 and with poor identity in sequence; however, some structural similarity is observed, which could suggest a similar role. The targets of PhbP2 or PhbP3 are not known, but some of them could be important to sustain normal growth under oxygen stress.

On the other hand, the higher rate of oxygen transfer in the cultures promotes the synthesis of P3HB, doubling of the polymer concentration in the three strains evaluated with respect to that obtained at a low oxygen transfer. Finally, it was observed that the molecular mass of the polymer synthesized by strain OP and OP-PhbP2^−^ decreased at the end of the culture, whereas in the case of the polymer obtained with the OP-PhbP3^−^ strain, it remains constant, a situation which could be related to the activity of the depolymerases. This suggests that a P3HB depolymerase could be a target of PhbP3.

## Figures and Tables

**Figure 1 polymers-16-02897-f001:**
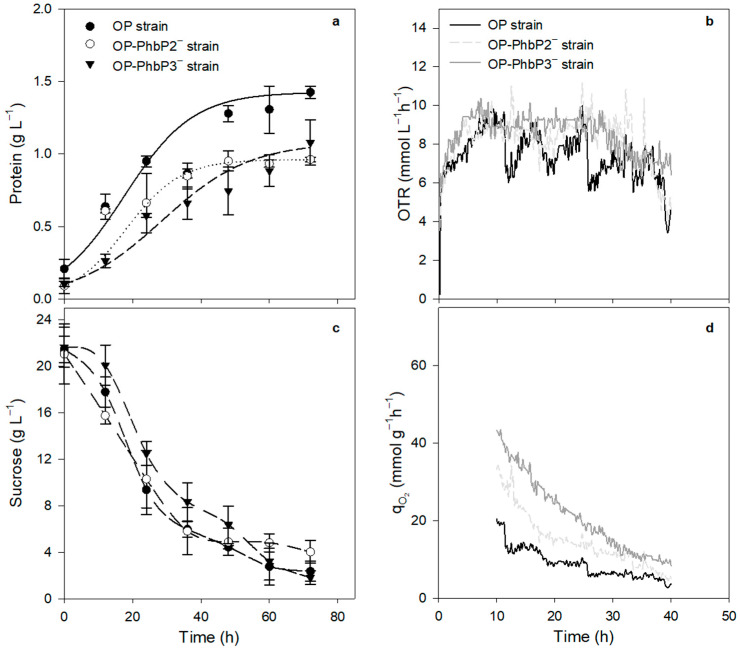
Bacterial growth kinetics (**a**), OTR evolution (**b**), sucrose (**c**) and qO_2_ evolution (**d**) for the strains cultured at 500 rpm.

**Figure 2 polymers-16-02897-f002:**
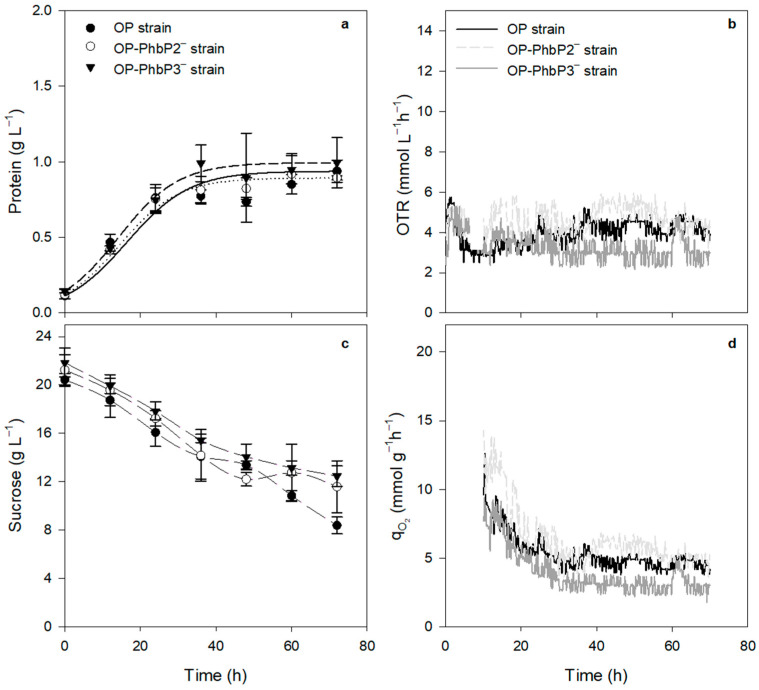
Bacterial growth kinetics (**a**), OTR evolution (**b**), sucrose (**c**) and qO_2_ evolution (**d**) for the strains cultured at 300 rpm.

**Figure 3 polymers-16-02897-f003:**
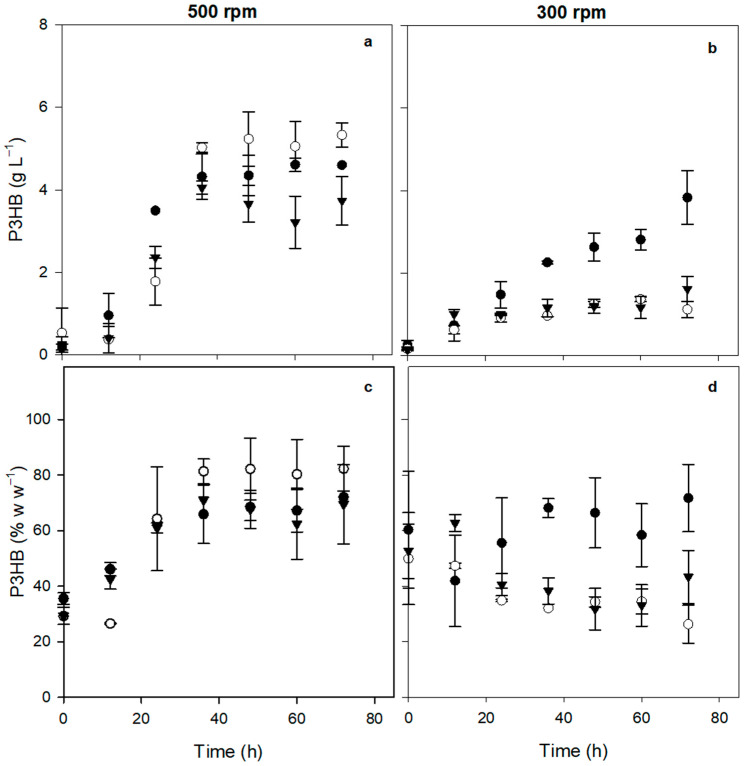
Evolution of the P3HB production (**a**,**b**) and intracellular accumulation (**c**,**d**) in cultures of three strains of *A. vinelandii* at different agitation rates (500 and 300 rpm). OP strain (black circles), OP-PhbP2^−^ (white circles) and OP-PhbP3^−^ strain (black triangles).

**Figure 4 polymers-16-02897-f004:**
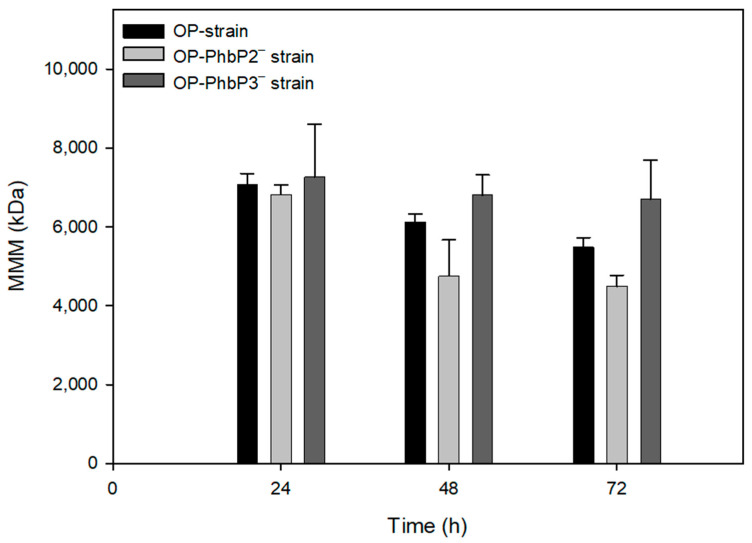
Mean molecular mass (MMM) of P3HB produced for the different strains evaluated in cultures of *A. vinelandii* developed at 500 rpm.

**Figure 5 polymers-16-02897-f005:**
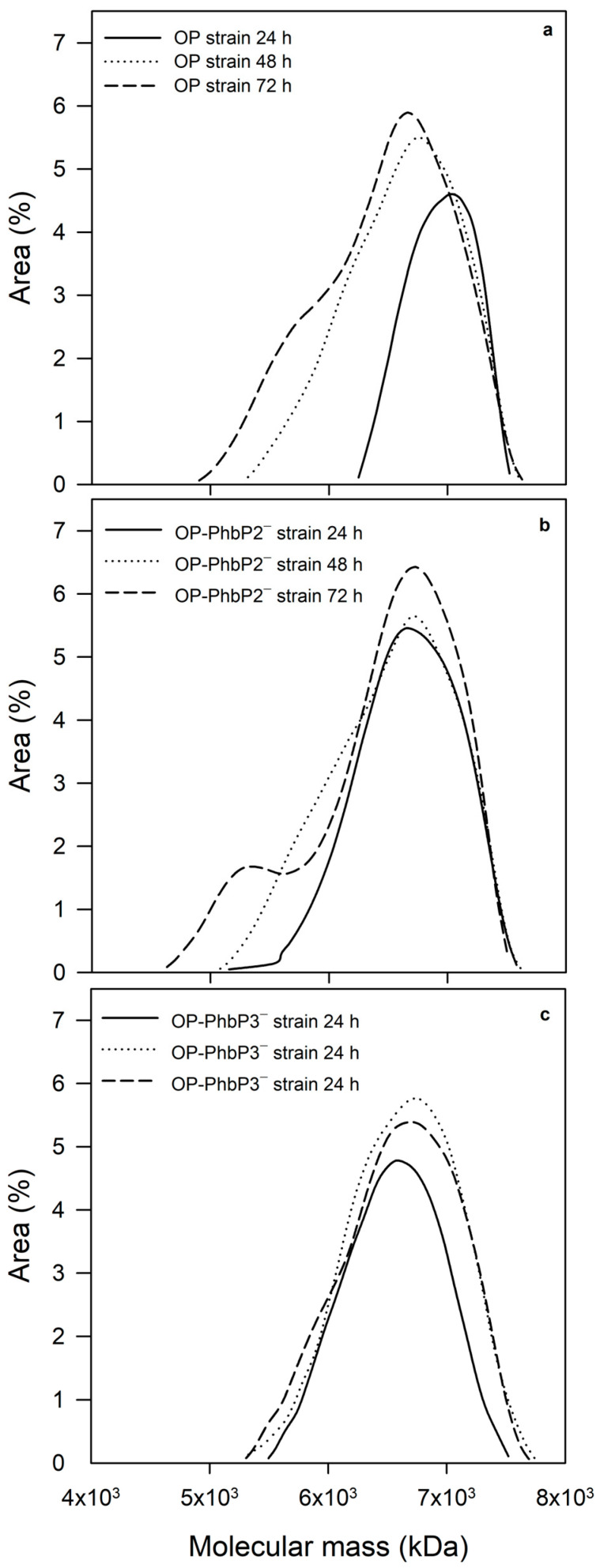
Distribution of molecular mass of P3HB produced by the OP (**a**), OP-PhbP2^−^ (**b**), and OP-PhbP3^−^ (**c**) strains in cultures performed at 500 rpm.

**Table 1 polymers-16-02897-t001:** Effect of agitation rate on the OTR, qO_2_, μ and final protein concentration in cultures of three strains of *A. vinelandii*.

Strain	Agitation Rate (rpm)	OTRₘₐₓ (mmol L^−1^ h^−1^)	qO_2_ ᵃ (mmoL g^−1^ h^−1^)	μ (h^−1^)	Cellular Protein (g L^−1^)
OP	500	8.30 ± 0.56	9.2 ± 1.0	0.080 ± 0.020	1.43 ± 0.04
	300	3.91 ± 0.90	6.9 ± 0.7	0.025 ± 0.001	0.94 ± 0.07
OP-PhbP2^−^	500	8.12 ± 0.24	25.5 ± 1.0	0.050 ± 0.010	0.96 ± 0.02
	300	4.94 ± 0.92	12.1 ± 3.8	0.025 ± 0.001	0.89 ± 0.01
OP-PhbP3^−^	500	8.94 ± 0.79	37.3 ± 1.6	0.040 ± 0.010	1.08 ± 0.15
	300	3.20 ± 1.00	6.9 ± 2.3	0.027 ± 0.001	0.99 ± 0.17

ᵃ measured at 12 h of cultivation.

**Table 2 polymers-16-02897-t002:** Effect of agitation rate on the P3HB production, accumulation of P3HB and P3HB volumetric productivity for the three strains evaluated of *A. vinelandii*.

Strain	Agitation Rate (rpm)	P3HBₘₐₓ (g L^−1^)	P3HBₘₐₓ (% on Dry Cell Weight)	Q_P3HB_ (g L^−1^ h^−1^)		
				24 h	48 h	72 h
OP	500	4.6 ± 0.5 (72 h)	72.0 ± 0.5	0.138 ± 0.003	0.087 ± 0.009	0.061 ± 0.001
	300	3.8 ± 0.6 (72 h)	69.0 ± 16.4	0.050 ± 0.009	0.049 ± 0.005	0.049 ± 0.008
OP-PhbP2^−^	500	5.3 ± 0.3 (72 h)	82.3 ± 8.1	0.052 ± 0.001	0.098 ± 0.001	0.067 ± 0.004
	300	1.4 ± 0.1 (60 h)	47.3 ± 1.0	0.031 ± 0.003	0.023 ± 0.001	0.013 ± 0.002
OP-PhbP3^−^	500	3.7 ± 0.6 (72 h)	71.0 ± 0.5	0.088 ± 0.003	0.071 ± 0.005	0.048 ± 0.006
	300	1.6 ± 0.3 (72 h)	62.7 ± 3.0	0.035 ± 0.001	0.022 ± 0.003	0.020 ± 0.004

**Table 3 polymers-16-02897-t003:** Relationship between the P3HB volumetric productivity and depolymerization rate of P3HB in cultures of OP, OP-PhbP2^−^, and OP-PhbP3^−^ strains at 500 rpm.

Strain	OTRₘₐₓ (mmol L^−1^ h^−1^)	Q_P3HB_ (g L^−1^ h^−1^)	Depolymerization Rate (Da h^−1^)
OP	8.30 ± 0.56	0.087 ± 0.009	15,800 ± 1500
OP-PhbP2^−^	8.12 ± 0.24	0.098 ± 0.001	24,500 ± 2500
OP-PhbP3^−^	8.94 ± 0.79	0.071 ± 0.005	3555 ± 136

## Data Availability

The original contributions presented in the study are included in the article, further inquiries can be directed to the corresponding author.

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
