# Peer review of "The Absence of Phasins PhbP2 and PhbP3 in Azotobacter vinelandii Determines the Growth and Poly-3-hydroxybutyrate Synthesis"

_polymers, 2024, doi:10.3390/polym16202897_

Round 1
Reviewer 1 Report
Comments and Suggestions for Authors
There is no numeration of lines in the text, and that is inconvenient for the work with the article. I recommend the authors to introduce such numeration. Abstract: I recommend to use phrase “affect the growth of polymer producer and polymer biosynthesis” instead of “affect the polymer metabolism” Sorry, but the phrase «The results reveal that under high OTRₘₐₓ conditions (between 8.1 and 8.9 mmol L⁻¹ h⁻¹ depending on the strain evaluated), the absence of phasins PhbP2 and PhbP3 resulted in a strong negative effect on the growth rate. In contrast, the rates of specific oxygen consumption increased in the case of cultures with strains where phasins were absent.» is unclear. In both cases the phasins are absent, but effect is different. Is this so? Section 2.1 Please add: - the wave length used for the determination of the optical density of cell suspensions and equipment used for this analysis, - the equipment used for the cell cultivation allowing to keep “200 rpm and 29 °C for 24 hours”. Section 2.3: Please give the equation of specific growth calculation instead of the reference to the name of other researchers like “García et al reported”. Section 2.4: The introduction of abbreviation “GPC” is not necessary, it was not used further in the text. The section says that the molecular weight is determined, and then the term “mean molecular mass” is introduced and the abbreviation MMM is introduced. It is necessary to bring everything into line. Section 3.1: please add the explanation of the difference between the OP-PhbP2 ̄ and OP-PhbP3 ̄ strains used in the work. Why did the difference only in the certain antibiotic resistance give so different results in the protein synthesis? Did you use some different plasmids? The main idea of the work is the comparison of the strains and a little information is given for their characterization. The information should be added to the Introduction. I recommend combining Tables 1 and 2, since the first three columns are the same, and the generalized table can be expanded to the width of the entire page. What time do the polymer accumulation results shown in Table 2 correspond to? I recommend that the authors check for themselves for which of the characteristics they discuss in the text, information on how they are calculated is not provided in the Materials and Methods section. In the study of the influence of phasins on the characteristics of the studied processes, the actual presence of phasins is not controlled in any way. Why? There is information only about the total cellular protein (Table 1), which is monitored to confirm cell growth. And where is the confirmation that phasins are synthesized at all and how do all the observed processes relate to their concentration? The analysis of these proteins could be done at least electrophoretically. Due to the fact that the presence of the discussed proteins and their concentrations were not controlled in any way, the authors' conclusion about how exactly these proteins (and no others synthesized by cells) affect the biosynthesis of polyhydroxybutyrates is not supported by anything. And if any other recombinant protein were synthesized in the same cells during the biosynthesis of a key biopolymer, would it also affect the synthesis of a biopolymer or not? It is known that recombinant proteins always increase the synthetic load on cells and become a priority in synthesis, while the synthesis of the main metabolites slows down. But this happens at the genetic level, and not due to the interactions of synthesized recombinant proteins and biopolymers. I think that the role of stafins has not been proven. It is necessary to supplement the article with controls and results. Why is there speculation about the depolymerization of synthesized polymers in cells, but the activity of depolymerases is not controlled in any way? I believe that the theory of depolymerase functioning is not confirmed by the results of the work. It is necessary to supplement the article with the control of these enzymes (their activity). Why does the work conclude that it is oxidative stress, and not mechanical stress, that causes the influence on the controlled parameters of the process? In order to unequivocally assert that the whole point is precisely in the concentration of available oxygen, it was worth not increasing the number of revolutions (mechanical action on cells), but using a system of bubbling the medium (forced air supply to the medium with control of the flow rate of supplied oxygen). Page 6, Section 3.2, paragraph 2. 5000 rpm is indicated instead of 500 rpm. Fugures 3c and 3d. A fundamental difference in the trends of biopolymer accumulation in cells at 500 and 300 rpm in time for two mutant strains is presented: at a lower mixing rate, the amount of polymer in cells only decreases. Why is this happening? And why is this observed only at 300 rpm? Why does Figure 4 below show data on molecular weight reduction only for cell culture at 500 rpm? These data are interpreted as the presence of depolymerization, which has not been proven! Therefore, Figure 6 is only an interpretation of the authors' own results, not the actual results, and can be removed from the article. Important. that the reason for the appearance of a polymer with a lower molecular weight may be that simply at later stages the polymer can be synthesized by cells with a lower molecular weight themselves, and hence there is a greater distribution of biopolymer molecules by fractions, and not because depolymerases work. In conclusion, it would be necessary to write where, in principle, these polymers with a higher or lower molecular weight are needed, how much they are in demand, since we are talking about publication in the journal Polymers, and discussion of the properties of polymers should remain in the first place.
Author Response
Reviewers' comments:
Manuscript Number: polymers-3198159
The absence of phasins PhbP2 and PhbP3 in Azotobacter vinelandii, determines the growth and poly-3-hydroxybutyrate synthesis
Reviewer 1
- There is no numeration of lines in the text, and that is inconvenient for the work with the article. I recommend the authors to introduce such numeration.
R: Thanks for the comments. The numeration was included in the new version of the manuscript.
- Abstract: I recommend to use phrase “affect the growth of polymer producer and polymer biosynthesis” instead of “affect the polymer metabolism” Sorry, but the phrase «The results reveal that under high OTRₘₐₓ conditions (between 8.1 and 8.9 mmol L⁻¹ h⁻¹ depending on the strain evaluated), the absence of phasins PhbP2 and PhbP3 resulted in a strong negative effect on the growth rate. In contrast, the rates of specific oxygen consumption increased in the case of cultures with strains where phasins were absent. » is unclear.
R: We appreciate the reviewer´s comments and in the new version of the manuscript we include the sentence “affect the growth of polymer producer and polymer biosynthesis” and the next paragraph “The results reveal that under high OTRₘₐₓ conditions (between 8.1 and 8.9 mmol L⁻¹ h⁻¹), the absence of phasins PhbP2 and PhbP3 resulted in a strong negative effect on the growth rate; in contrast, the rates of specific oxygen consumption increased in the both cases (Page 1, lines: 15-20).
- In both cases, the phasins are absent, but effect is different. Is this so?
R: In terms of the growth rate and the specific oxygen consumption rate, the effect is the same, since in both cases, there is a decrease in the specific growth rate; wheras, the oxygen consumption rate increases with respect to the values determined in the cultures with OP strain. This behavior is described and discussed on page 5 lines 224- 227, and in table 1.
- Section 2.1 Please add: - the wave length used for the determination of the optical density of cell suspensions and equipment used for this analysis, - the equipment used for the cell cultivation allowing to keep “200 rpm and 29 °C for 24 hours”. R: In the new version of the manuscript information regarding the wavelength and equipment was included (Page 3, lines 109, 110).
- Section 2.3: Please give the equation of specific growth calculation instead of the reference to the name of other researchers like “García et al reported”.
R: Thanks for the correction, in the new version of the manuscript, the equation was included (Page 4, 5, line 133-136).
- Section 2.4: The introduction of abbreviation “GPC” is not necessary, it was not used further in the text. The section says that the molecular weight is determined, and then the term “mean molecular mass” is introduced and the abbreviation MMM is introduced. It is necessary to bring everything into line.
R: Thanks for the comments, this was corrected throughout the manuscript (Page 5, line 160 and 172).
- Section 3.1: please add the explanation of the difference between the OP-PhbP2 ̄ and OP-PhbP3 ̄ strains used in the work. Why did the difference only in the certain antibiotic resistance give so different results in the protein synthesis? Did you use some different plasmids?
R: The reviewer is right, a better explanation about the characteristics of the two mutant strains used in this work is needed. Three different phasin proteins (PhbP1, PhbP2 and PhbP3) have been identified in A. vinelandii, but only the most abundant phasin PhbP1 has been studied, having a role in the determination of the size a number of P3HB granules (Moreno et al., 2019). About PhbP2, that belongs to the same family as PhbP1, and PhbP3, which represents a new family of phasins, little is known. To start understanding their physiological role, in this work two mutants were studied. Strain OP- PhbP2⁻, which is a strain that has the phbP2 gene inactivated, and strain OP- PhbP3⁻ that has the phbP3 gene inactivated. Then, their responses to two different OTR conditions, one of them considered as representing oxidative stress, were studied.
This information was not sufficiently explained in the previous version; therefore, in this new version of manuscript, a brief description of the phasin proteins and the corresponding mutants was included in section 3.1 and 3.2 (Page 5, Lines 186-208; 212-217), as suggested by the reviewer, but we also included in section 2.1 (Materials and methods; Page 2, 3; lines 86-100), a better description of the genetic modifications present in each mutant to indicate that two different phasins, named PhbP2 and PhbP3, were inactivated in mutants OP-PhbP2 ̄ and OP-PhbP3 ̄ respectively, and explaining what phasin was inactivated in each mutant. We also included in section 1 (Introduction) a brief description of the two phasin proteins of A. vinelandii that were studied in this work (Page 2, lines 64-68).
- The main idea of the work is the comparison of the strains and a little information is given for their characterization. The information should be added to the Introduction.
R: We agree, the previous manuscript had no information about the mutant strains and the genes inactivated. We incorporated this information in Materials and methods (Page 3, Lines 90-104) and also in the Introduction (Page 2, Lines 63-67).
- I recommend combining Tables 1 and 2, since the first three columns are the same, and the generalized table can be expanded to the width of the entire page. What time do the polymer accumulation results shown in Table 2 correspond to? I recommend that the authors check for themselves for which of the characteristics they discuss in the text, information on how they are calculated is not provided in the Materials and Methods section.
R: Thanks for the reviewer comments. In the new version of the manuscript, Table 2 was modified deleting the OTR values and including the P3HB volumetric productivity calculated during the cultivation. Thus, Table 2 shows values related to P3HB production. The form to calculate the P3HB volumetric productivity is shown in the new version of the manuscript (Page 4, line 137-140).
- In the study of the influence of phasins on the characteristics of the studied processes, the actual presence of phasins is not controlled in any way. Why? There is information only about the total cellular protein (Table 1), which is monitored to confirm cell growth. And where is the confirmation that phasins are synthesized at all and how do all the observed processes relate to their concentration? The analysis of these proteins could be done at least electrophoretically.
R: The reviewer is right, the manuscript had little information about the kind of mutations introduced and this is confusing. Strains OP-PhbP2- and OP-PhbP3- ̄ are mutant derivatives of strain OP containing gene inactivations of two different phasin genes, phbP2 and phbP3 respectively. The inactivations consisted in introducing antibiotic resistance cassettes within the coding sequences of phbP2 or phbP3, producing truncated proteins. Therefore, no functional PhbP2 protein was produced in strain OP-PhbP2- ̄and no PhbP3 protein was produced in OP-PhbP3-. These two gene insertions were constructed in vitro and were introduced into the chromosome of strain OP by double-crossover recombination, replacing the wild type allele with the truncated gene. The gene replacements in the chromosome were confirmed by PCR. This information is now provided in the new version of Materials and methods (Lines 86-101) and Results (Page 5, lines 186-208).
- Due to the fact that the presence of the discussed proteins and their concentrations were not controlled in any way, the authors' conclusion about how exactly these proteins (and no others synthesized by cells) affect the biosynthesis of polyhydroxybutyrates is not supported by anything. And if any other recombinant protein were synthesized in the same cells during the biosynthesis of a key biopolymer, would it also affect the synthesis of a biopolymer or not? It is known that recombinant proteins always increase the synthetic load on cells and become a priority in synthesis, while the synthesis of the main metabolites slows down. But this happens at the genetic level, and not due to the interactions of synthesized recombinant proteins and biopolymers. I think that the role of stafins has not been proven. It is necessary to supplement the article with controls and results. Why is there speculation about the depolymerization of synthesized polymers in cells, but the activity of depolymerases is not controlled in any way? I believe that the theory of depolymerase functioning is not confirmed by the results of the work.
- R: We agree, no definitive conclusion about the role of phasins PhbP2 or PhbP3 can be established; however, the observed phenotypes can be attributed to the absence of these proteins, because the gene inactivations introduced specifically eliminated their expression. No recombinant proteins were expressed in these mutants, they only lack the corresponding phasin. The observed phenotypes are induced by the absence of the phasin, although indirect effects induced by this lack of phasin cannot be discarded. The suggestion about the possible interaction of the phasin with P3HB depolymerases is based on previous reports showing that some phasins control the activity of depolymerizing enzymes (Handrick et al., 2004; Kuchta et al., 2007), and also on our results showing that the changes in the molecular weight distributions that occur during stationary phase, induced by the P3HB depolymerase PhbZ1 (Adaya et al., 2018; Millán et al., 2016), are modified in the mutants, and also on the difference in depolymerase activities that we measured in strain OP and the mutants.
- It is necessary to supplement the article with the control of these enzymes (their activity). Why does the work conclude that it is oxidative stress, and not mechanical stress, that causes the influence on the controlled parameters of the process?
R: Thanks for the reviewer's comments. In order to support the hypothesis that phasin PhbP3 could be regulating depolymerase activity, we have included in the new version of the article the values of the depolymerase activities of the different strains evaluated. Results shows the absence of depolymersase activity in the cultures with the OP-PhbP3- strain at the end of the culture (72 h) unlike what is observed in the cultures with the OP and OP-PhbP2- strains, where detects an activity. This was included in the new version of the manuscript (Page 13, line 449-453).
- In order to unequivocally assert that the whole point is precisely in the concentration of available oxygen, it was worth not increasing the number of revolutions (mechanical action on cells), but using a system of bubbling the medium (forced air supply to the medium with control of the flow rate of supplied oxygen). Page 6,
R: Thanks for the reviewer's comments. It is a good cultivation strategy to increase the flow or bubbling of the bioreactor, without increasing the agitation speed. However, it is known that A. vinelandii cells walls exhibits a high resistance to disruption by mechanical agitation and the susceptibility depends on the cell morphology [Peña et al., 2000]. Therefore, it seems unlikely that cells of A. vinelandii could be damaged by the mechanical stress generated in the bioreactor at 500 rpm This paragraph was included in te new version of the manuscript (Page 7, line 278-280).
- Section 3.2, paragraph 2. 5000 rpm is indicated instead of 500 rpm.
R: This correction was done in the new version of the manuscript (page 9, line 320).
- Figures 3c and 3d. A fundamental difference in the trends of biopolymer accumulation in cells at 500 and 300 rpm in time for two mutant strains is presented: at a lower mixing rate, the amount of polymer in cells only decreases. Why is this happening? And why is this observed only at 300 rpm?
- R: Thanks for the comments. It is correct that at 300 rpm in the mutant strains there is a decrease in PHB accumulation according to the culture time. Although we do not have an explanation for this, it is possible that in the mutant strains there is a slight consumption of PHB from the beginning of cultivation, which is not necessarily reflected in the concentration of the polymer. This paragraph was included in the new version of the manuscript (Page 9, line 343-349).
- Why does Figure 4 below show data on molecular weight reduction only for cell culture at 500 rpm? These data are interpreted as the presence of depolymerization, which has not been proven! Therefore, Figure 6 is only an interpretation of the authors' own results, not the actual results, and can be removed from the article.
R: Thanks for the comment from the reviewer. The new version includes data on depolymerase activity, which supports the idea that the decrease in molecular weight in the OP and P2- strain is related to greater depolymerase activity (Page 13, line 431-435).
- Important. that the reason for the appearance of a polymer with a lower molecular weight may be that simply at later stages the polymer can be synthesized by cells with a lower molecular weight themselves, and hence there is a greater distribution of biopolymer molecules by fractions, and not because depolymerases work.
R: Thanks for the comment. In the new version of the manuscript In the new version of the manuscript, information on the depolymerase activity is included to explain the drop in the molecular weight of PHB synthesized by the OP and OP-PhbP2- P2 strains (Page 13, line 431-435).
- In conclusion, it would be necessary to write where, in principle, these polymers with a higher or lower molecular weight are needed, how much they are in demand, since we are talking about publication in the journal Polymers, and discussion of the properties of polymers should remain in the first place.
R: In the new version of the manuscript, a brief description of the molecular mass and properties of the polymer was included (Page 11, line 417-423).
Reviewer 2 Report
Comments and Suggestions for Authors
The manuscript was not good enough in terms of language and writing. Moreover, the paper has the scope to an improvement.
1. Overall, there were confusion between phasins PhbP2 and PhbP3 and strain PhbP2 and PhbP3, could be specify or make link between them.
2. In abstract: ‘It was observed, at high OTRₘₐₓ that the absence of phasin PhbP3 harms the P3HB synthesis, finding a decrease of 30 % in the polymer production at the end of cultivation; however, the molecular weight of the polymer remained constant during the course of cultivation.’ – this sentence is too long. Divide it to make the outcome more clearly.
3. In abstract: ‘In summary, it can be concluded that both growth and polymer synthesis were affected when phasin PhbP3 was absent from the P3HB granule, especially at high OTRₘₐₓ.’ – I am not clear on this statement.
4. Aim of the work need to rewrite: it must be clear and easy to follow.
5. Objectives and aim of the works was not clear and not linked with the introduction and literature review?
6. Give the reference of Eq. 1.
7. What are the significance of (1 − 𝑋𝑂₂ 𝑖𝑛 − 𝑋𝐶𝑂₂ in/1−𝑋𝑂₂ 𝑜𝑢𝑡 − 𝑋𝐶𝑂₂ 𝑜𝑢𝑡) in Eq. 1?
8. What was the cause of decrease of MMM over time?
9. What was the QP3HB?
10. Rewrite the conclusion more clearly and from the findings from the research. were not interlinked and not up to mark to make proper conclusion.
11. In conclusion: ‘Overall, our results have demonstrated, for the first time, that the absence of phasin PhbP3 and PhbP2 causes a significant decrease in specific growth rate in cultures of the mutant strains conducted at high OTR.’ – I did not find this concluding statement evaluated in the experimental study.
Comments on the Quality of English LanguageExtensive editing of English language required.
Author Response
The manuscript was not good enough in terms of language and writing. Moreover, the paper has the scope to an improvement.
- Overall, there were confusion between phasins PhbP2 and PhbP3 and strain PhbP2 and PhbP3, could be specify or make link between them.
R:The denomination PhbP2 and PhbP3 refer to the phasin proteins; whereas, OP-PhbP2- or OP-PhbP3- refer to the corresponding mutants. We revised this nomenclature in the manuscript (Page 5, lines 215, 216).
- In abstract: ‘It was observed, at high OTRₘₐₓ that the absence of phasin PhbP3 harms the P3HB synthesis, finding a decrease of 30 % in the polymer production at the end of cultivation; however, the molecular weight of the polymer remained constant during the course of cultivation.’ – this sentence is too long. Divide it to make the outcome more clearly.
- R: Thanks of the comment. We change the sentence for “It was observed that at high OTR, the absence of PhbP3 affected the production of P3HB, decreasing by 30 % at the end of cultivation. In contrast, the molecular weight remained constant over time” (Page 1, line 23-27).
- In abstract: ‘In summary, it can be concluded that both growth and polymer synthesis were affected when phasin PhbP3 was absent from the P3HB granule, especially at high OTRₘₐₓ.’ – I am not clear on this statement.
R: In the new version of the manuscript this sentence was changed by
“In summary, the absence of phasin PhbP3 from the P3HB granule significantly impacted the growth rate and polymer synthesis, particularly at high maximum oxygen transfer rates (OTRₘₐₓ)” (Page 1, lines 25-27).
- Aim of the work need to rewrite: it must be clear and easy to follow.
R: In the new version the objective was rewritten by
“The aim of the present study was to evaluate the effect of inactivating phasins PhbP2 or PhbP3 on the growth rate, production and molecular mass of P3HB in cultures at low and high oxygen transfer rates (OTR)………”(Page 2, line 79-83).
- Objectives and aim of the works was not clear and not linked with the introduction and literature review?
R: Thanks for the comments. In the new version the objective was rewritten.
- Give the reference of Eq. 1.
- R: In the new version of the manuscript the reference is indicated (Page 3, 4 line 123-126).
- What are the significance of (1 − ??₂ ?? − ???₂ in/1−??₂ ??? − ???₂ ???) in Eq. 1?
R: Effectively, as commented by the reviewer, the term described in eq. 1 is not significant. We revised the data and in the new version of the manuscript, we modified the equation and did not include this term (Page 3, line 124).
- What was the cause of decrease of MMM over time?
R: Thanks for the comment. The mainly reason it’s due to the depolymerization activity. In order to support this hypothesis we have included in the new version of the article the values of the depolymerase activities of the different strains evaluated. Results shows the absence of depolymersase activity in the cultures with the OP-PhbP3- strain at the end of the culture (72 h) unlike what is observed in the cultures with the OP and OP-PhbP2- strains, where detects an activity. This was included in the new version of the manuscript (Page 13 lines 449-458).
- What was the QP3HB?
R: This term refers to the volumetric production of PHB. In the new version of the manuscript the definition is included on page 4 line lines 137-138.
- Rewrite the conclusion more clearly and from the findings from the research. were not interlinked and not up to mark to make proper conclusion.
R: Thanks to the reviewer for his comment. In the new version of the manuscript, we have changed the conclusion (Page 14, lines 514-524).
“Overall, our results demonstrate for the first time that the absence of phasins PhbP3 and PhbP2 leads to a significant reduction in the specific growth rate of mutant strains cultured under high OTR conditions (8.3-8.9 mmol /l h). This behavior could be related to the protective role of these proteins against the effect of oxidative stress to which the cells are subjected in the cultures at high oxygen transfer. On the other hand, the increment in the rate of oxygen transfer in the culture promotes the synthesis of P3HB, reaching the doubling of the polymer concentration in the three strains evaluated, with respect to that obtained at a low oxygen transfer. Finally, it was observed that the molecular mass of the polymer synthesized by strain OP and OP-PhbP2- decreases at the end of the culture; whereas, in the case of the polymer obtained with the OP-PhbP3- strain remain constant, a situation which is related to the activity of the depolymerases”
- In conclusion: ‘Overall, our results have demonstrated, for the first time, that the absence of phasin PhbP3 and PhbP2 causes a significant decrease in specific growth rate in cultures of the mutant strains conducted at high OTR.’ – I did not find this concluding statement evaluated in the experimental study.
R: Thanks for the comment
Table 1 shows how the specific growth rate of the OP-PhbP2- and OP-PhbP3- strains decreases around 50 % with respect to the value calculated with the OP strain in the cultures developed at an agitation of 500 rpm (OTR = 8.3-8.9 mmol /l h).
Reviewer 3 Report
Comments and Suggestions for Authors
Attached file.

Author Response
In this manuscript, two mutants of Azotobacter vinelandii presenting interruption in two Phasin genes were evaluated for cell growth, oxygen consumption, P3HB production, depolymerization and molecular weight. The work is quite relevant and give new interesting informations about the roles of Phasins, proteins that have other roles beyond the PHA granules stabilization. The manuscript is ready for publication after minor modifications.
- Since CO2 in the outlet gas was measured, it will be very interesting present the data of specific rate of CO2 production and the respiratory quotient (RQ).
R: We revised with more detail the data of CO2 and it is not possible to present the
evolution of CO 2 produced and hence calculate the RQ. According to that, in the new version of the manuscript, the equation to calculate the OTR has been modified.
- Line 69: “…it has been reported in that the increase in the agitation rate in…” Suggestion: “…it has been reported that the increase in the agitation rate in…”
R: Thanks for the comments. In the new version, we incorporate the suggestion (Page 2, line 72).
- Line 203: “qO” Suggestion “qO2”.
R: Thanks for the comments. In the new version are remarked the term “qO2” (Page 7, line 268).
Reviewer 4 Report
Comments and Suggestions for Authors
The manuscript by Zapata and co-authors explores the role(s) of two phasins in PHA accumulation under different aeration conditions.
Comments
L 81-85. If the knock-out strains were constructed in the present study it is necessary to include details on the procedures used for recombineering. If not, please clarify the text and include a reference.
L 148. The study would be significantly more comprehensive and in-depth with a comparative analysis of the sequences and 3D models of the PhbP2 and PhbP3 phasins from A. vinelandii. How different are these protein sequences and 3D model structures? Please include this analysis in the results and discussion section.
L 152-158. Are the observed differences statistically significant? Please include this in the text and figures.
L 164. Please state in which interval µ values were calculated.
L 233-236. Are the observed differences statistically significant? Please include this in the text and figures.
L 244-245. It is unclear why different time points were used to calculate the %P3HBmax, is it knock-out dependent? Please include this discussion in the text.
L 271-273. Are the observed differences statistically significant? Please include this in the text and figures.
L 326. Please add an equation to calculate QP3HB in the methods section.
L 328. How was the depolymerization rate calculated? Please include this in the methodology section.
L 328-332. Please include a figure showing the volumetric productivity values for each strain during cultivation.
L 354-355. Please make clear that you are proposing distinct functions for each phasin studied, and what these functions are.
Author Response
The manuscript by Zapata and co-authors explores the role(s) of two phasins in PHA accumulation under different aeration conditions.
Comments
- L 81-85. If the knockout strains were constructed in the present study, it is necessary to include details on the procedures used for recombineering. If not, please clarify the text and include a reference.
R: The reviewer is right, the manuscript had little information about the construction of the mutants. Strains OP-PhbP2- and OP-PhbP3 ̄ are mutant derivatives of strain OP containing gene inactivations of two different phasin genes, phbP2 and phbP3 respectively. The inactivations consisted in introducing antibiotic resistance cassettes within the coding sequences of phbP2 or phbP3, producing truncated proteins. This information is now provided in the new version of Materials and methods (Page 2 lines 86-101).
- L 148. The study would be significantly more comprehensive and in-depth with a comparative analysis of the sequences and 3D models of the PhbP2 and PhbP3 phasins from A. vinelandii. How different are these protein sequences and 3D model structures? Please include this analysis in the results and discussion section.
R: We agree with the reviewer, the comparative analysis of the sequences of PhbP2 and PhbP3 is needed. It was incorporated in the Results and discussion section (Page 5, lines 186-208).
- L 152-158. Are the observed differences statistically significant? Please include this in the text and figures.
R: According the ANOVA analysis the difference are significant. A p value of 0.008 was used in the analysis.
- L 164. Please state in which interval µ values were calculated.
R: Thanks for the comments. The interval considered was from 0 to 24 hours. We indicated the interval in the new version (Page 3, 4, lines 133-136).
- L 233-236. Are the observed differences statistically significant? Please include this in the text and figures.
R: Statistical analysis was performed and included in the new version of the manuscript. In general, no significant differences were found in the P3HB accumulation in the different strains evaluated (Page 9, line 321, 323).
- L 244-245. It is unclear why different time points were used to calculate the % P3HBmax, is it knock-out dependent? Please include this discussion in the text.
R: It is correct that the maximum accumulation of PHB occurs at different cultivation times. The above is related to the oxygen transfer conditions and the type of strain used. In general, the maximum accumulation values are reached at the beginning of the stationary phase of growth (36-48 h); however, in the case of the mutant strains (OP-PhbP2- and OP-PhbP3 ̄ ) grown at 300 rpm, the behavior is different, with the maximum accumulation observed at the beginning of the culture 12-24 h. We still don't know why this behavior occurs. In the new version of the manuscript a brief description of this phenomenon is included on the pages 9 and 10, lines 343, 349.
- L 271-273. Are the observed differences statistically significant? Please include this in the text and figures.
R: According the ANOVA analysis there are significant differences in the MMM of the polymer produced by the OP and OP-PhbP2- in the range of 24 to 72 h; whereas no significant differences were found in the MMM of the PHB produced by the OP-PhbP3- in the same period of cultivation, this was included in the new version of the manuscript (Page 11, lines 404-408).
- L 326. Please add an equation to calculate QP3HB in the methods section.
R: Thanks for the comments, we include the equation in the new version (Page 4, lines 137-140)
- L 328. How was the depolymerization rate calculated? Please include this in the methodology section.
R: The depolymerization rate was calculated taking into account the change in the mean molecular mass (MMM) in the range of 24 to 72 h, using the following equation:
Depolymerization rate = MMM 72h – MMM 24 h/ 48 h
This paragraph was included in the methodology (Page 4, 5; Lines, 177-182).
- L 328-332. Please include a figure showing the volumetric productivity values for each strain during cultivation.
R: In the new version of the manuscript, the P3HB volumetric productivity of each strain to three times of cultivation was included (Table 1).
- L 354-355. Please make clear that you are proposing distinct functions for each phasin studied, and what these functions are.
R: Thanks for the comments. We included a brief description about the functions of phasins in the new version of the manuscript (Page 2, line 48,49; 53,54; 63-67; Page 5, line 186-188).
Round 2
Reviewer 1 Report
Comments and Suggestions for Authors
The new section 3.1 contains only published results. There are no new data. Since then the information should be moved from Results and Discussion to the Introduction.
I recommend the authors to perform the correct equation instead of given as equation #2 for the calculation of the μ. For this calculation the equation of the following type “μ = ….” should be given. Please, completely modify the text at lines 128-130.
There is some unknown symbol (ç ) in Table 2, line with “3.8 ± 0.6ç (72 h)”. It should be removed.
Line 394: please, check the word “inactiation”.
Figure 6 is very simple, and I recommend to replace it by the Table with the same data.
References: all references should be performed by the same way (##8,9,25,28,33) and in accordance with the template (recommendations) of the Journal. The DOI should be added to all references. See lines 512 and 515 – there are wrong numbers of the references.
Author Response
The new section 3.1 contains only published results. There are no new data. Since then the information should be moved from Results and Discussion to the Introduction.
R: Thanks for the comments. The information was moved from Results and Discussion to the Introduction in the new version of the manuscript. Page 2, lines 62-81.
I recommend the authors to perform the correct equation instead of given as equation #2 for the calculation of the μ. For this calculation the equation of the following type “μ = ….” should be given. Please, completely modify the text at lines 128-130.
R: Thanks for the comments. The equation 2 was corrected in the new version of the manuscript. Page 3, lines 144, 147
There is some unknown symbol (ç ) in Table 2, line with “3.8 ± 0.6ç (72 h)”. It should be removed.
R: The correction was done in the new version of the manuscript. Table 2
Line 394: please, check the word “inactiation”.
R: The correction was done in the new version of the manuscript. Line 392.
Figure 6 is very simple, and I recommend to replace it by the Table with the same data.
R: In the new version of the manuscript Figure 6 was replaced by table 3.
6) References: all references should be performed by the same way (##8,9,25,28,33) and in accordance with the template (recommendations) of the Journal. The DOI should be added to all references. See lines 512 and 515 – there are wrong numbers of the references.
R: The references were performed using the same format and including the DOI.
Round 3
Reviewer 1 Report
Comments and Suggestions for Authors
I am satisfied with the changes introduced by the authors to the manuscript based on my recommendations.